# Verification of Key Target Molecules for Intramuscular Fat Deposition and Screening of SNP Sites in Sheep from Small-Tail Han Sheep Breed and Its Cross with Suffolk

**DOI:** 10.3390/ijms25052951

**Published:** 2024-03-03

**Authors:** Lingjuan Fu, Jinping Shi, Quanlu Meng, Zhixiong Tang, Ting Liu, Quanwei Zhang, Shuru Cheng

**Affiliations:** 1College of Animal Science and Technology, Gansu Agricultural University, Lanzhou 730070, China; flj18419143575@163.com (L.F.); shijinpingyunfei@163.com (J.S.); mengquanlu2021@163.com (Q.M.); tzx18153613730@163.com (Z.T.); liut@gsau.edu.cn (T.L.); 2College of Life Science and Technology, Gansu Agricultural University, Lanzhou 730070, China; zhangqw@gsau.edu.cn

**Keywords:** sheep, meat quality, IMF, RNA-Seq, SNP

## Abstract

Intramuscular fat (IMF) is vital for meat tenderness and juiciness. This study aims to explore the IMF deposition mechanism and the related molecular markers in sheep. Two populations, Small-tail Han Sheep (STH) and STH × Suffolk (SFK) F_1_ (SFK × STH), were used as the research object. Histological staining techniques compared the differences in the longissimus dorsi muscle among populations. A combination of transcriptome sequencing and biological information analysis screened and identified IMF-related target genes. Further, sequencing technology was employed to detect SNP loci of target genes to evaluate their potential as genetic markers. Histological staining revealed that the muscle fiber gap in the SFK × STH F_1_ was larger and the IMF content was higher. Transcriptome analysis revealed that *PIK3R1* and *PPARA* were candidate genes. Histological experiments revealed that the expressions of *PIK3R1* mRNA and *PPARA* mRNA were lower in SFK × STH F_1_ compared with the STH. Meanwhile, PIK3R1 and PPARA proteins were located in intramuscular adipocytes and co-located with the lipid metabolism marker molecule (FASN). SNP locus analysis revealed a mutation site in exon 7 of the *PIK3R*1 gene, which served as a potential genetic marker for IMF deposition. This study’s findings will provide a new direction for meat quality breeding in sheep.

## 1. Introduction

The sheep meat industry has always played a vital role in animal husbandry, especially in the northern regions of China, where there is a high demand for lamb. Stable growth in China’s sheep meat industry will increase the income level of farmers and herdsmen, ensuring the economic stability of ethnic minority areas. Gansu’s dry climate, with sparse precipitation, leads to a high evaporation rate. The fertile soil and grassland ecology are suitable for the growth of the sheep industry. The nationally renowned “Jingyuan Lamb” and “Linxia Hand Grab Meat” have become unique brands of Gansu lamb [1,2,3]. Recently, Small-tail Han sheep (STH) have been the primary breed in the pastoral area of Gansu in recent years. This breed is characterized by early maturity, fast growth, tolerance to rough feeding, and good adaptability [4]. In order to fully utilize the growth and reproductive performance of STH while improving meat quality, Suffolk (SFK) sheep with high meat performance have been introduced for hybrid improvement. High-level reproductive performance was maintained by increasing the feed conversion rate and meat production rate, which further resulted in the shortening of the feeding cycle and acceleration of the growth speed, resulting in the production of high-quality green mutton products in large quantities to increase sheep breeding income [5,6,7]. The lamb quality directly affects consumers’ choices [8]. Intramuscular fat (IMF) is a key factor determining meat quality, affecting meat tenderness, flavor, and juiciness [9,10,11]. Therefore, it is particularly important to search for target molecules related to IMF deposition in sheep and improve their meat quality at the molecular level.

However, since IMF belongs to quantitative traits and a trait is jointly controlled by micro multi genes and major genes [12,13], conventional phenotypic selection was used to obtain ideal traits. Although the ideal traits improved meat quality, they also consumed a lot of manpower and material resources. Moreover, the selection of traits was unstable and prone to segregation. The understanding of the genetic mechanism of IMF deposition can help improve IMF content through molecular breeding. In recent years, increased research aims to elucidate the molecular mechanism of IMF traits with the development of high-throughput sequencing and bioinformatics.

Transcriptome analysis revealed that *FAM134B*, *FAS*, *PPARγ*, and *HSL* can be used as candidate genes for IMF deposition [14,15,16,17]. Xueying Zhang et al. [18] conducted a joint analysis between the fatty acid profile and transcriptional profile of Tan sheep, indicating that *ADIPOQ*, *FABP4*, *PLIN1*, *PPARGC1A*, and *SLC2A1* accelerated IMF deposition by positively regulating the metabolism of saturated and monounsaturated fatty acids. By using chromatin sequencing (ATAC seq) and RNA sequencing (RNA seq), Zhong Xu [19] identified chromatin regions and key genes that affect IMF content in Xidu black pig breeds. The results revealed that *PVALB*, *THRSP*, *HOXA9*, *EEPD1*, *HOXA10*, and *PDE4B* may be related to fat deposition. Further, the Protein–Protein Interaction (PPI) prediction analysis indicated that the *PVALB* gene was the highest hub gene. Susan K. Duckett [20] demonstrated that bta-miR-122 may be related to IMF deposition by conducting miRNA transcriptome sequencing on bovine longissimus dorsi muscle tissue. By transcriptome sequencing analysis of chicken pectoral and abdominal muscles, Na Luo [21] reported that the IMF deposition in pectoral muscles was associated with pyruvate and citrate metabolism through genes such as *GAPDH*, *LDHA*, *GPX1*, and *GBE1.* IMF deposition in abdominal muscles was associated with acetyl CoA and glycerol metabolism through *FABP1*, *ELOVL6*, *SCD*, and *ADIPOQ*. These studies have significantly improved our understanding of the characteristics of IMF deposition. However, it is still unclear whether the genes mentioned above are the key genes affecting IMF deposition. The molecular mechanism of IMF deposition remains unresolved, and novel methods are required to unravel the genes and pathways regulating IMF content. Single nucleotide polymorphism (SNP), as an ideal genetic marker, has a wide range of applications in analyzing genetic diversity variety identification and analyzing the relationship between genotype and phenotype. The molecular marker technology to obtain genes or SNPs or molecular marker assisted selection can improve meat quality in a relatively efficient and feasible method [22,23].

In this study, the histological structure of the longissimus dorsi muscle of STH and the SFK×STH F_1_ population was compared. Transcriptome sequencing on the longissimus dorsi muscle tissue was conducted; key candidate genes for IMF deposition were screened and identified, and their functions were explored. Meanwhile, the candidate gene of the SNP loci was analyzed in detail to elucidate the possible reasons for different IMF deposition abilities in sheep. The key mutation sites related to IMF deposition genes were explored, providing molecular markers for sheep meat quality breeding.

## 2. Results

### 2.1. Comparison of IMF Content in the Longest Muscle Tissue of Different Groups of Sheep

The STH and SFK × STH F_1_ population was used as the research object to observe and compare the differences in the histological structure and IMF content of the longissimus dorsi muscle between the two populations (Figure 1). The results showed a tight connection between multiple muscle fibers in the STH population, with small gaps between muscle fibers and visible lipid droplet structures in the muscle bundle gaps. The gaps between muscle fibers in the SFK × STH F_1_ population indicated the place where the muscle fibers structure was loose. Lipid droplets were observed between muscle fibers as well as between muscle bundles, indicating that the SFK × STH F_1_ group had greater IMF deposits and IMF content. Lipid droplet deposition in the muscle fiber gaps of the STH population was observed via Oil red O staining, showing a concentrated distribution and almost no lipid droplets in other parts. The SFK × STH F_1_ population had a large number of lipid droplets dispersed across muscle fibers and intercellular spaces, along with a wide distribution area, indicating that the SFK × STH F_1_ population had a higher IMF content. The results of statistical analysis showed that there was a statistically significant difference in the IMF content between the two populations (Appendix A) (*p* < 0.01).

### 2.2. Differential Gene Identification and Screening Based on GO Analysis

RNA seq analysis was performed on the longissimus dorsi muscle tissue of the SFK × STH F_1_ (high IMF) and the STH (low IMF) populations to screen for DEGs. The potential function of the DEGs in lipid metabolism was studied using GO terms and KEGG pathway enrichment analysis. The results of GO analysis showed that the biological processes involved in sheep lipid metabolism (Figure 2) were mainly enriched in adipocyte metabolism, adipocyte differentiation, and lipid reactions. Here, 26 DEGs were identified, of which *C3*, *ELOVL7*, *PPARA*, *ABHD2*, and *PPARD* were enriched in more than five lipid metabolism-related biological processes. Except for C3 upregulated expression, all other DEGs were downregulated. The molecular function (Figure 3) was mainly enriched in lipid binding, lipase activity, and long-chain fatty acid binding, resulting in the identification of 22 DEGs, of which only *PPARD* participated in all three molecular functions simultaneously. Additionally, KEGG analysis showed (Figure 4) that DEGs were mainly related to the regulation of fat degradation, MAPK signaling pathway, and PPAR signaling pathway in adipocytes, including 40 DEGs, where 16 were involved in multiple pathways, five were downregulated, and 11 were upregulated. These results indicated that these genes were probably involved in IMF deposition in sheep.

### 2.3. Gene Co-Expression

The above-mentioned genes selected for the GO and KEGG metabolic pathways were merged and deduplicated to identify the core genes, and the Pearson correlation coefficient (cor) between the genes was calculated. The genes with a cor value >0.6 were selected to construct a gene co-expression network diagram to identify the hub genes (Figure 5). A total of 23 nodes were identified in this network, of which *PIK3R1*, *PPARA*, *SNAI2*, *ABHD2*, *CERM*, *ELOVL7*, *FGFR4*, *FOXO1*, and *MSTN* nodes had strong connections. *PIK3R1* and *PPARA,* with the strongest connections, were selected as the hub genes.

### 2.4. Functional Analysis of Candidate Genes Related to IMF

The expression of *PIK3R1* and *PPARA* mRNA in the longissimus dorsi muscle tissue of different sheep populations was analyzed using qRT-PCR to verify the reliability of transcriptome sequencing (expression of transcriptome sequencing genes expressed using FPKM values) results and explore the functions of *PIK3R1* and *PPARA* genes in IMF deposition (Figure 6). The results showed that the expression of *PIK3R1* mRNA in the STH population was found to be significantly higher compared to those in the SFK × STH F_1_ population (*p* < 0.01). The expression of *PPARA* mRNA in the STH was found to be significantly higher than that in the SFK × STH F_1_ population (*p* < 0.05), consistent with the transcriptome sequencing gene expression pattern, indicating that the DEGs identified from RNA Seq in this study were reliable. The expression of *PIK3R1* mRNA in the SFK × STH F_1_ population with high intramuscular fat content was also lower than that in the STH, indicating that *PIK3R1* and *PPARA* gene expression probably had partial inhibitory effect on intramuscular fat deposition.

### 2.5. Immunohistochemistry

The expression and localization of PIK3R1 and PPARA in the longissimus dorsi muscle tissue were observed via IHC staining to further investigate the role of PIK3R1 and PPARA in IMF deposition (Figure 7). The results showed that PIK3R1 and PPARA proteins were positively expressed in the fibrous space of the longissimus dorsi muscle, and the positive staining degree of PIK3R1 in the STH was significantly higher than that in the SFK × STH F_1_ population (*p* < 0.01). The positive staining degree of PPARA was significantly greater than that in the SFK × STH F_1_ population (*p* < 0.05). Also, there were no positive staining reactions for PIK3R1 and PPARA in the negative control group, indicating that PIK3R1 and PPARA inhibited the deposition of IMF by affecting lipid metabolism in IMF cells.

### 2.6. Co-Positioning

IF staining was used to observe the co-localization of PIK3R1 protein and PPARA protein with lipid metabolism marker molecule FASN to further clarify the mechanism of PIK3R1 and PPARA in IMF deposition. The results showed that PIK3R1 and PPARA proteins were positively expressed in the intramuscular adipocytes in the fiber gap of the longissimus dorsi muscle and co-localized with FASN in the cytoplasm (Figure 8). The values for the Pearson correlation coefficients (Rr) of PIK3R1 and FASN were 0.97 and 0.96, respectively. The overlap coefficients (R) between the STH and the SFK × STH F_1_ population was 0.96. The results showed almost complete overlap between two proteins, indicating that PIK3R1 was closely related to lipid metabolism. The values for the Rr of PPARA and FASN between the two populations were 0.72 and 0.81, respectively, and the R were 0.67 and 0.75, respectively. Not all PPARA signals overlapped with FASN signals, indicating that although PPARA was related to lipid metabolism, it was probably also associated with other molecules at different cellular locations.

### 2.7. PIK3R1 Single Nucleotide Polymorphism Site

The co-localization results showed that *PIK3R1* was a key target molecule for IMF deposition in sheep. Next, single nucleotide polymorphism site detection was performed to identify key mutation sites to investigate the causes of different intramuscular fat deposition in sheep caused by *PIK3R1*. The results showed that there was a missense site in the exon 7 region of the *PIK3R1* gene, C1146T (Appendix A). The mutation site mutated from base C to base T at g.1146, causing a glutamic acid mutation encoding selenocysteine. There were three genotypes, i.e., CC, CT, and TT. The genetic diversity analysis of the SNP site of the *PIK3R1* gene (Table 1) showed that CT was the dominant genotype and C was the dominant allele. The PIC classification criteria revealed that the mutation at this site in the *PIK3R1* gene belonged to a highly polymorphic site (*p* > 0.5).

### 2.8. PIK3R1 Protein Physical and Chemical Parameters

Table 2 shows that the molecular weight, isoelectric point, instability index, total number of negatively charged amino acids, and total average hydrophilicity of the protein had changed before and after the mutation, along with the physical and chemical properties of the protein. These changes were found to be associated with changes in the protein’s function, resulting in different fat deposition abilities.

### 2.9. PIK3R1 Protein Structure Prediction

The results presented in Table 3 show that the secondary structure of proteins α Spiral, angle, and random curl changed before and after the mutation. The prediction of the secondary protein structure showed that there were changes in the secondary protein structure before and after the mutation site. Figure 9 and Figure 10 show that the tertiary structure of the protein underwent spatial changes before and after mutation, leading to changes in the protein’s function.

## 3. Discussion

Meat is one of the primary sources of high-quality protein in human diets. With improving living standards and dietary structure, there is an increasing preference for meat products from herbivorous animals. The quantity and distribution of intramuscular fat—also known as marbled fat—is ideal for enhancing the flavor and palatability of meat [24]. Consumers favor Japanese Wagyu for its ability to deposit fat between muscle fibers, forming a symmetrical marble-shaped snowflake meat [25,26] This study focuses on the STH and the SFK × STH F_1_ with different meat qualities. H&E staining and oil red O staining were used to observe the histological characteristics and lipid droplet distribution of the longissimus dorsi muscle. The staining revealed the presence of vacuolar lipid droplet structures in the muscle fiber gaps of the sheep’s longissimus dorsi muscle. The muscle fiber gaps of the SFK × STH F_1_ were wider and looser than STH, and most lipid droplets were observed in the SFK × STH F_1_ population. Further, the IMF content of SFK × STH F_1_ was higher than that of the STH, indicating that muscle fiber gaps can be filled by adipocytes, providing environmental support for IMF deposition. Previously, IMF deposition in cattle has been reported to depend on muscle fiber gaps. The more IMF deposited in muscle fiber gaps, the higher the quality of beef and lamb [27,28], consistent with the results of this study.

IMF content significantly impacts meat quality and flavor. However, it has not been well studied due to the complex formation mechanism of the IMF. Comparing transcriptome studies of individuals with different phenotypic traits is helpful in analyzing animal IMF traits [29]. This study identified the main biological processes involved in IMF deposition in sheep through transcriptome analysis: adipocyte metabolism, adipocyte differentiation, and lipid response. The main metabolic pathways include adipocyte lipid degradation regulation, MAPK signaling pathway, and PPAR signaling pathway, among which *PIK3R1* and *PPARA* are key target molecules for IMF deposition. The PI3K/Akt pathway is associated with adipocyte differentiation; inhibiting PI3K/Akt activation inhibits adipogenesis and differentiation [30,31,32]. The *PIK3R1* gene can affect fat deposition through the PI3K/Akt signaling pathway, further confirming the accuracy of the GO terminology and KEGG pathway genes related to IMF deposition reported in this study.

*PIK3R1* is a key gene regulating animal growth and glucose and lipid metabolism. Several studies have demonstrated that the *PIK3R1* gene plays a vital role in the molecular mechanism of animal fat deposition. The *PIK3R1* mutation—where Arginine is replaced by tryptophan—has been reported to lead to disorders in human lipid metabolism. The mutation results in a loss of the hydroxyl group binding ability of PIK3 to the substrate, which leads to partial fat malnutrition [33]. This gene is a marker gene related to human lipid metabolism. Currently, there is not much information on its role in mammalian growth, development, and fat metabolism. This study analyzed the gene expression profiles of *PIK3R1* in the longissimus dorsi muscle tissue of two populations and found the *PIK3R1* mRNA expression level in SFK × STH F_1_ was extremely significantly lower than that in the STH; the trend was the reverse of that of the IMF content. IHC and IF staining also demonstrated that the PIK3R1 protein was located in the IMF cells in the muscle fiber gap, almost entirely co-localizing with the lipid metabolism marker molecule FASN. This suggests that PIK3R1 may inhibit intramuscular fat deposition by altering the direction of lipid metabolism in the body.

The *PPARA* gene is a fat-specific gene that can enhance fatty acid oxidation when activated. Knocking out *PPARA* increases fat production in the white adipose tissue of mice [28]. This study used qRT-PCR technology to detect the expression level of *PPARA* mRNA in the longissimus dorsi muscle tissue of different populations to investigate the specific mechanism of the *PPARA* gene in the process of IMF deposition in sheep. The *PPARA* mRNA of SFK × STH F_1_ was significantly lower than that of the STH, contrary to the oil red O staining results. The PPARA protein expression in the longissimus dorsi muscle tissue was observed through IHC and IF staining to clarify its mechanism of action further. PPARA protein was found to be localized in intramuscular adipocytes between muscle fibers and co-localized with the lipid metabolism marker FASN, suggesting that PPARA might inhibit IMF deposition by promoting fatty acid oxidation.

In summary, the findings of this manuscript suggest that *PIK3R1* and *PPARA* are key target molecules for IMF deposition. The different IMF deposition abilities of the sheep population were investigated by analyzing the SNP sites of the candidate genes. Mutations in the exon 7 region of the *PIK3R1* gene changed the amino acid sequence and protein secondary structure, affecting its function. Sun Guangrong [34] analyzed the association between *PIK3R1* gene polymorphism and fat deposition and tail type in sheep. They also detected the mutation site. They further reported that *PIK3R1* had 1 SNPs loci, one of which strongly correlated with the back fat thickness of SFK hybrid sheep. There was also a significant correlation between *PIK3R1* gene haplotype and the back fat thickness and IMF content of SFK hybrid sheep. The size of genetic variation within a population is expressed through the number of effective alleles [35]. The distribution of alleles within the population, as well as their heterozygosity, is also a reference number for genetic variation in the population. In this study, the *PIK3R1* gene in the SFK × STH F_1_ has a small degree of variation, high genetic information content, and high heritability. This indicates the potential for genetic selection and can be used as a molecular marker for sheep IMF breeding.

## 4. Materials and Methods

### 4.1. Animal Welfare

All samples were collected in alignment with the ethical standards approved by the Animal Welfare Committee of the School of Animal Science and Technology, Gansu Agricultural University (GSAU-AEW-2017-0308).

### 4.2. Sample Collection and Preparation

The experiment was conducted on the STH, and the SFK × STH F_1_ population from Linxia Prefecture, Gansu Province, was selected as the research object. Here, 25 STH and 30 SFK × STH F_1_ were chosen randomly. Blood samples were collected from the sheep from the jugular vein and stored in an EDTA anticoagulant blood collection tube in dry ice before being stored at −80 °C for future use. Six sheep (aged 6 months) with similar health and good physical condition were randomly selected from each of the two groups, and the muscle samples of the longissimus dorsi muscle tissue were collected, packaged, and placed in cryopreservation tubes and sampling tubes, respectively, in liquid nitrogen and fixed with 4% paraformaldehyde. They were transported back to the laboratory on the same day for subsequent experiments. The specific information of the 6-month-old sheep is shown in the Table 4.

### 4.3. Hematoxylin-Eosin (H&E) Staining Technique

Different groups of longissimus dorsi tissues were fixed using 4% paraformaldehyde, and paraffin sections (thickness 3–4 µm) were prepared following the method of Wen Lei et al. [36]). The specimens were baked at 60 °C for 4 h, dewaxed using xylene, dehydrated with gradient alcohol, and stained with hematoxylin for nuclei, followed by treatment with HCl differentiation solution. Next, the specimens were stained with eosin for cytoplasm, dehydrated with gradient alcohol, treated with xylene, sealed with neutral gum, and naturally dried. The images were captured and visualized using an Olympus BX53M microscope(Olympus Corporation, Tokyo, Japan).

### 4.4. Oil Red O Staining

The longissimus dorsi muscle tissue was fixed using 4% paraformaldehyde, followed by wrapping with OCT. Then, the tissue was placed in a freezer for the solidification of the OCT. After solidification, the specimens were cut into 8 µm-thick slices. The frozen slices were reheated and dried, fixed, washed with water, and air-dried. Next, the slices were dipped in oil red dye solution in the dark and then re-dyed with hematoxylin, followed by washing with pure water. The slices were then treated with a differentiation solution, returned to blue, and soaked in tap water. The staining effect on the slices was microscopically examined, sealed with glycerol–gelatin sealing agent, and air dried before collecting images. The proportion of positive staining area was measured and analyzed using the Image J software (v1.8.0.345), and the data were plotted using GraphPad Prism 7.0 after statistical processing.

### 4.5. Total RNA Preparation, cDNA Library, and Sequencing

Under sterile conditions, 100 mg of each group of longissimus dorsi tissue was taken and ground with liquid nitrogen. Total RNA was extracted from the longissimus dorsi tissue following the instructions of the total RNA extraction kit (Nanjing Novozan, Chinadfe, Nanjing, China). The RNA’s quality and quantity were evaluated using spectrophotometers (Pulton, MO, USA) and bioanalyzers (Sangta Clara, CA, USA). Next, magnetic beads containing Oligo (dT) were used to enrich the mRNA of the longissimus dorsi muscle. One strand of cDNA was randomly synthesized using mRNA as a template, followed by two-strand cDNA synthesis. After purification of the double-strand cDNA, end repair and sequencing were performed. The fragment size was selected using AMPure XP beads (USA Beckman, Brea, CA, USA) (length 200–500 bp). Finally, PCR amplification was performed to construct the cDNA library. Finally, the Illumina HiSeqTM 4000 platform (Illumina, San Diego, CA, USA) was used for sequencing. Transcriptome sequencing was completed by Guangzhou Saizhe Biological Company (Guangzhou, China).

### 4.6. Transcriptome Analysis and Differential Expression Gene Identification

The clean reads obtained from sequencing were strictly filtered. The reads containing sequencing joint reads with N content > 10% and low-quality reads were removed to obtain high-quality clean reads (alkali base with a mass value of SQ ≤ 20 accounted for >50% of the entire read). Next, the clean reads were compared to the sheep reference genome (Oar. v 4.0) using transcriptome data comparison software Bowtie2 (v2.2.9) and TopHat2 (v2.1.1) [37,38] to obtain the comparison results for each sample. The FPKM algorithm was used for standardization of the expression of transcripts [39], with FPKM ≥ 0.1 [40] using TopHat2 (v2.1.1) [37,38] to obtain STH as a control, and with FDR < 0.01 and |log2FC| > 1 as the standard [41]. Differentially expressed genes (DEGs) were screened to compare the gene expression between different populations. The genes were annotated, and their functions were obtained based on GO and KEGG databases. Finally, the GO function and KEGG Pathway were identified in significant DEGs with *p*-values ≤ 0.05 for subsequent analysis.

### 4.7. Real-Time Fluorescence Quantitative PCR (qRT-PCR)

The sample and method for RNA extraction were consistent with RNA sequencing. Total RNA (500 ng) was used to perform reverse transcription following the instructions of the Evo MLV reverse transcription kit (Acore, Nanjing, China). The cDNA sample was diluted after reverse transcription to 80 ng/µL for qRT-PCR using *GAPDH* as the internal reference gene. Gene-specific primers were designed using Primer 3.0 (Appendix A). The PCR reaction had a total volume of 20 μL including 0.5 μL upstream and 0.5 μL downstream primers, cDNA (1 μL), 2× SYBR Green Pro Taq HS Premium 10 μL, and 8 μL RNase free water. The reaction procedure was as follows: pre-denaturation at 95 °C for 30 s, denaturation at 95 °C for 5 s, annealing at 60 °C for 30 s; then, 45 cycles of 95 °C for 10 s, 65 °C for 60 s, 97 °C for 1 s, and 37 °C for 30 s. The cycle was repeated thrice for each sample. The relative expression of mRNA was calculated using the 2^−∆∆ct^ method, repeated thrice for each experimental group.

### 4.8. Immunohistochemistry Staining (IHC)

The paraffin slices were baked in a 65 °C oven for 3 h, dewaxed, and dehydrated, followed by antigen repairing using sodium citrate buffer. Next, 3% H_2_O_2_ was added dropwise to cover the tissue entirely, and it was placed in a wet box for 10 min. The tissue staining SP (Streptomyces avidin peroxidase) kit A solution (sealed solution) was added and incubated for 30 min. The slices were kept under overnight incubation with 1:300 diluted rabbit anti-PIK3R1 and 1:60 diluted mouse anti-PPARA in a 4 °C wet box (negative control had PBS instead of the primary antibody). Next, B solution was added dropwise and incubated at 37 °C for 30 min, followed by the addition of C solution dropwise and incubation at 37 °C for 30 min. After washing thrice with PBS, a 3′- diaminobenzidine (DAB) reagent kit was used for staining, and the stained sections were placed in tap water to terminate DAB staining. Hematoxylin was added and allowed to stand for 1–3 min. Then, the slices were rinsed with running water for 10 min, immersed in HCl/alcohol differentiation solution for 1–2 s, and rinsed with water again for 15 min to allow the solution to turn blue. Next, the slices were dehydrated using gradient alcohol, treated with xylene, and sealed with gum. The staining results were visualized using an Olympus DP71 optical microscope and imaging system. The positive signal area ratio (APSAP) of the image was analyzed using Image j software (v1.8.0.345) and visualized using a bar chart.

### 4.9. Immunofluorescence Staining of Tissues (IF)

After dewaxing and dehydration of the paraffin sections, the tissue sections were subjected to antigen repair by adding sodium citrate antigen repair buffer using the microwave heating method. After sealing at room temperature for 30 min with 5% donkey serum, 1:300 rabbit anti-FASN antibody (Boason, Beijing, China) was added, followed by 1:300 rabbit anti-PIK3R1 antibody (Boason, Beijing, China) and 1:60 diluted mouse anti-PPARA (Boason, Beijing, China) and kept in overnight incubation at 4 °C. The sections were washed thrice with PBS. FITC-labeled goat anti-rabbit IgG secondary antibody (1:300) was added dropwise (Boason, Beijing, China), followed by incubation at room temperature in the dark for 1 h. Then, the sections were washed thrice with PBS, followed by the addition of 0.5 mg/mL DAPI staining in the dark for 3–5 min. Finally, a small amount of anti-fluorescence quenching sealing agent was added to seal the film. LSM800 laser confocal scanning microscope (CARL ZEISS, Oberkochen, Germany) was used for film visualization and image analysis, and the co-localization coefficient was calculated using Fiji 64.0 software.

### 4.10. Single Nucleotide Polymorphism Analysis and Bioinformatics Analysis of PIK3R1

The frozen blood samples were melted at a gradient temperature, and genomic DNA was extracted from the blood samples following the operating instructions of the blood DNA extraction kit. The purity and concentration of DNA with purity (1.8–2.0) were used as pure DNA for subsequent amplification (Appendix A ). The amplification primers were designed using Primer 3.0 (Same as 4.7). The PCR amplification products were detected by 1.2% agarose gel electrophoresis, and the PCR amplification products were sent to Guangzhou Jinweizhi Biotechnology Co., Ltd., Guangzhou, China for Sanger sequencing of PCR amplification products: An online website was used to investigate the physicochemical properties of PIK3R1 protein (http://web.expasy.org/protparam/, accessed on 9 November 2023) and the Secondary Structure (http://npsapbil.ibcp.fr/cgibin/npsa_automat.pl?page=npsa_sopma.html, accessed on 9 November 2023) and Three-level Structure (https://swissmodel.expasy.org/, accessed on 9 November 2023) to conduct predictive analysis.

### 4.11. Statistical Analysis

Statistical analysis was performed on qRT-PCR data using SPSS 22.0. The data were analyzed using a *t*-test and one-way ANOVA method, and the results were expressed as mean ± standard deviation (X ± SD). Statistical charts were drawn using GraphPad Prism 9.0 and Adobe Illustrator. *p* < 0.05 indicated significant differences, while *p* < 0.01 indicated extremely significant differences. The sequencing data were organized and summarized using an Excel spreadsheet, followed by calculation of the genotype frequencies of different mutation sites using the Excel spreadsheet, including expected heterozygosity (He), effective allele number (Ne), polymorphism information content (PIC), and testing whether they were in a Hardy Weinberg equilibrium state.

## 5. Conclusions

In summary, this study reports marker molecules for the selection of an IMF trait in sheep. Potential candidate genes (*PIK3R1* and *PPARA*) and metabolic pathways (adipocyte metabolism, regulation of adipose degradation within adipocytes, MAPK signaling pathway, and PPAR signaling pathway) related to fat deposition were detected via transcriptome sequencing. SNP locus analysis confirmed that *PIK3R1* might be responsible for differential IMF deposition ability of the sheep population in this study. Further, *PIK3R1* can be used as an IMF deposition marker for further research, providing valuable information for genetically improving meat quality.

## Figures and Tables

**Figure 1 ijms-25-02951-f001:**
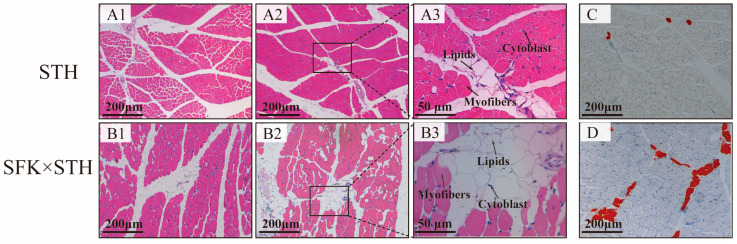
Cross-section staining of the longissimus dorsi muscle (Note: (**A1,B1**,**A2**,**B2**,**A3**,**B3**) show H&E staining, (**A3**) shows local amplification of (**A2**), and (**B3**) shows local amplification of (**B2**); the box represents the enlarged areaand, and the arrows indicate cytoblast, myofibers, and lipids. (**C**,**D**) is oil red O staining, The red area represents lipids).

**Figure 2 ijms-25-02951-f002:**
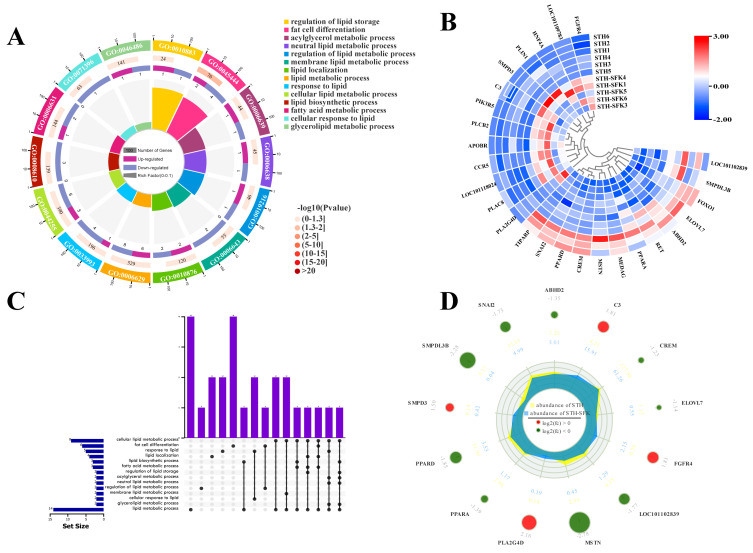
Screening of lipid metabolism-related differential genes based on GO BP terminology (**A**) Enrichment circle diagram; (**B**) Heat map; (**C**) Upset Venn diagram; (**D**) Radar chart.

**Figure 3 ijms-25-02951-f003:**
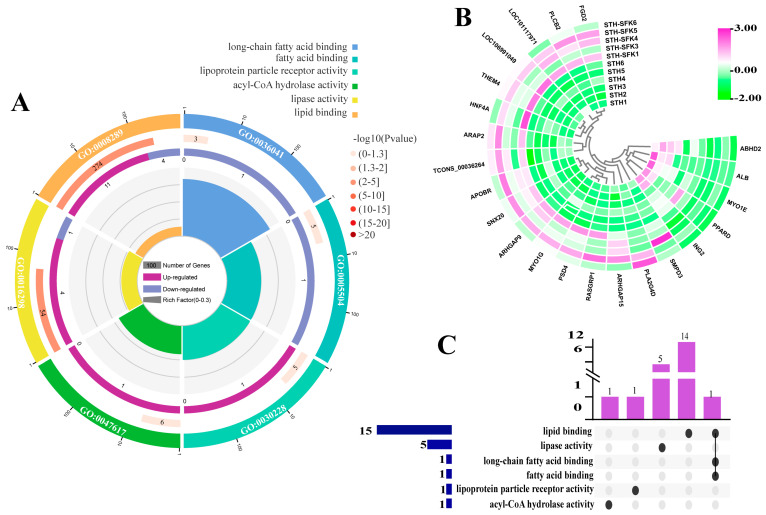
Screening of lipid metabolism-related differential genes based on GO MF terminology (**A**) Enrichment circle diagram; (**B**) Heat map; (**C**) Upset Venn diagram.

**Figure 4 ijms-25-02951-f004:**
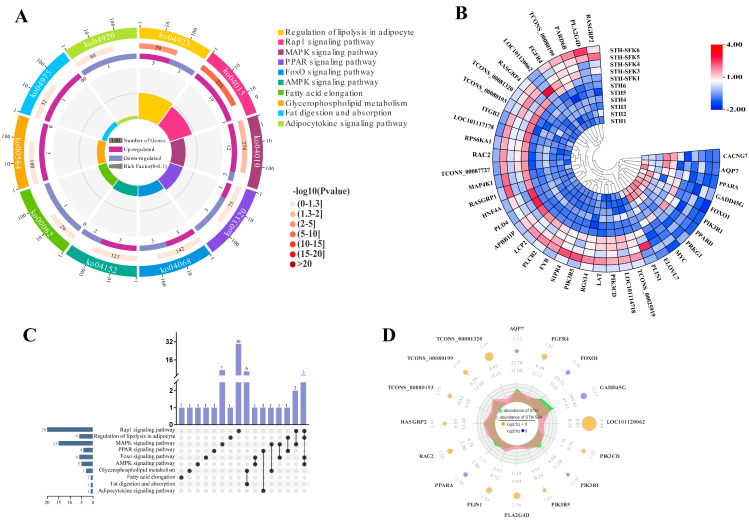
Screening of lipid metabolism-related differential genes based on KEGG. (**A**) Enrichment circle diagram; (**B**) Heat map; (**C**) Upset Venn diagram; (**D**) Radar chart.

**Figure 5 ijms-25-02951-f005:**
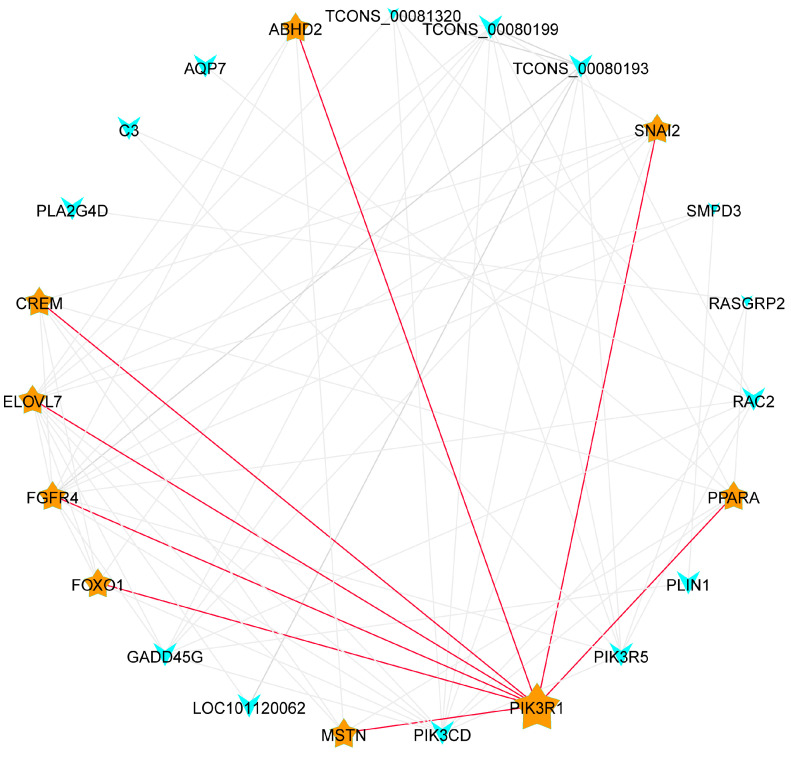
Gene co-expression network map (Note: Nodes represent genes, straight lines represent the regulatory relationship of genes, and node size represents the number of genes interacting with them, which is expressed by connectivity).

**Figure 6 ijms-25-02951-f006:**
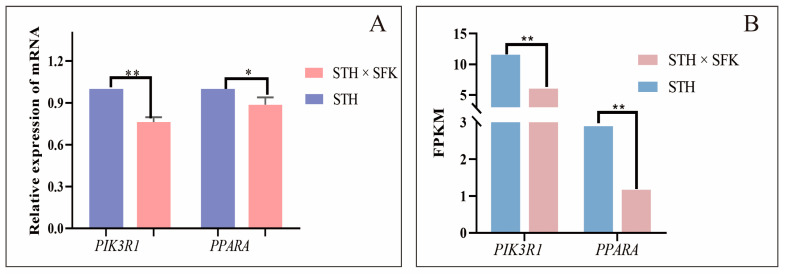
Relative expression of PIK3R1 and PPARA mRNA in the longissimus dorsi muscle tissue (Note: (**A**) shows the relative expression of mRNA; (**B**) shows the gene expression. The bar chart ** indicates significant differences (*p* < 0.01), and * indicates significant differences (*p* < 0.05).

**Figure 7 ijms-25-02951-f007:**
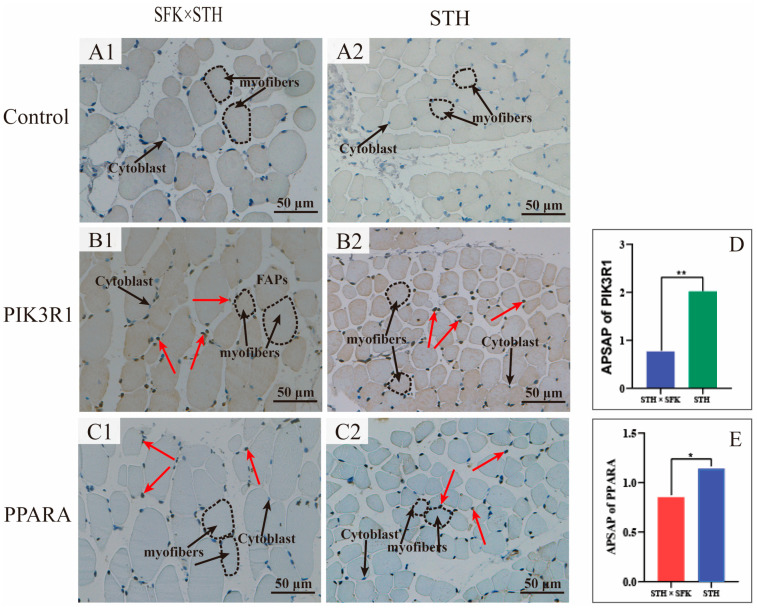
Immunohistochemical staining for the expression and localization of PIK3R1 and PPARA proteins in the longissimus dorsi muscle tissue (**A1**,**A2**). Negative control group; (**B1**,**B2**). PIK3R1 protein expression distribution; (**C1**,**C2**). PPARA protein expression distribution. Cytoblast. Epicell nucleus; myofiber. The red arrow indicates positive expression; the circle indicates one myofiber, the black arrow indicates cytoblast and myofibers.(**D**,**E**). Protein positive expression histogram. The bar chart ** indicates significant differences (*p* < 0.01), and * indicates significant differences (*p* < 0.05).

**Figure 8 ijms-25-02951-f008:**
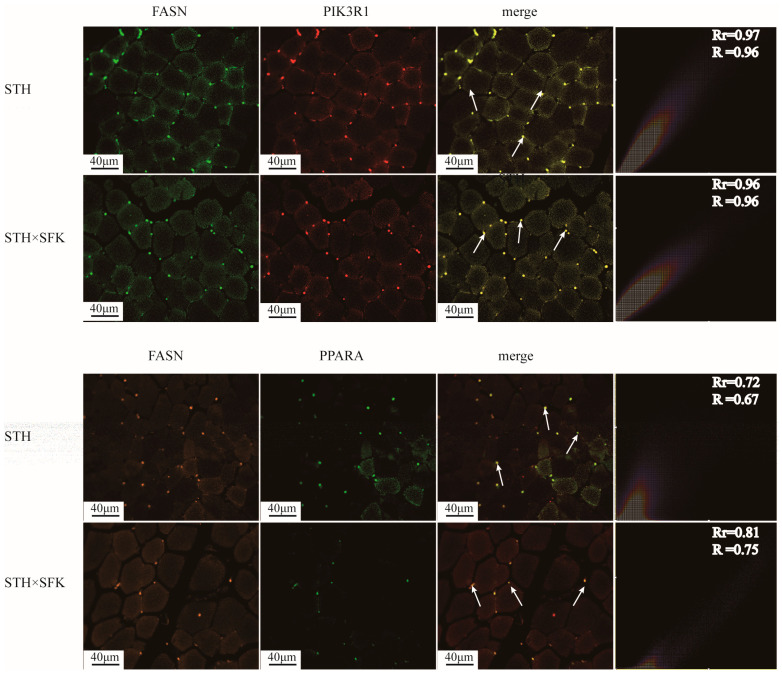
Immunofluorescence staining for the expression and localization of PIK3R1 protein in the longissimus dorsi muscle tissue. Upper part of the group diagram: Green fluorescence represents FASN protein, red fluorescence represents PIK3R1, and merge column shows this fluorescence color overlap; Upper part of the group diagram: Orange fluorescence represents FASN protein, green fluorescence represents PPARA, and merge column shows the overlap of these fluorescence colors; The white arrow represents the overlap of two proteins.

**Figure 9 ijms-25-02951-f009:**
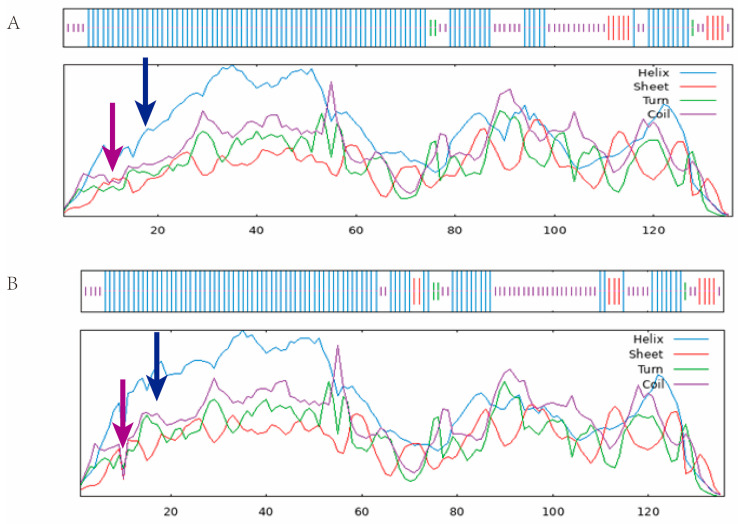
The secondary structure of the protein before and after the mutation (Note: (**A**) Before the mutation; (**B**) After the mutation, the blue arrow shows the Alpha helix; the purple arrow shows the Random coil).

**Figure 10 ijms-25-02951-f010:**
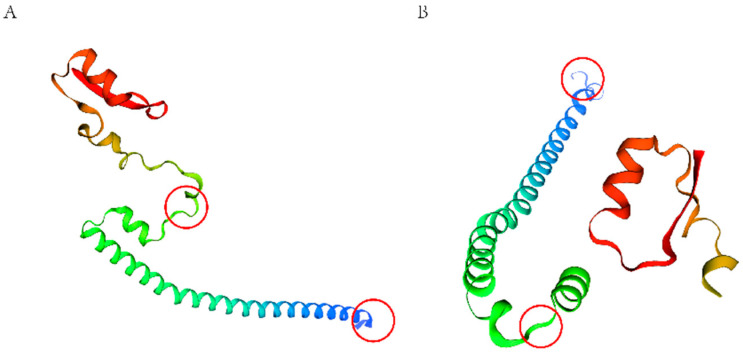
The tertiary structure of the protein before and after mutation of PIK3R1. (**A**) Before the mutation; (**B**) after the mutation. the red circles indicates the location where the structure has changed.

**Table 1 ijms-25-02951-t001:** Population genetic diversity analysis of *PIK3R1* SNPs loci.

SNPs	Groups	Number	Genotype			Gene		Ho	He	Ne	PIC	X2 (*P*)
C1164T			CC	CT	TT	C	T					
	STH	25	0.24 (6)	0.48 (12)	0.28 (7)	0.48	0.52	0.50	0.50	1.99	0.63	0.04 (0.98)
	SFK × STH	30	0.40 (12)	0.47 (14)	0.13 (4)	0.64	0.36	0.47	0.53	1.85	0.65	0.01 (0.99)
	All	55	0.33 (18)	0.47 (26)	0.20 (11)	0.57	0.43	0.49	0.51	1.96	0.63	0.10 (0.95)

Note: The SNP locus genotypes are wild-type, heterozygous mutant, and homozygous mutant genotypes from left to right; Ho: homozygosity; He: heterozygosity; Ne: number of effective alleles; PIC: polymorphism information content; X2 (*P*): chi square (*p*-value); PIC < 0.25 indicates low polymorphism, 0.25 ≤ PIC < 0.50 indicates moderate polymorphism, and PIC ≥ 0.50 indicates high polymorphism.

**Table 2 ijms-25-02951-t002:** Physicochemical properties of proteins before and after PIK3R1.

Physical and Chemical Parameters	Before Mutation	After Mutation
Molecular weight	16,443.35	16,464.27
Theoretical pI	6.02	6.39
Total number of negatively charged residues (Asp + Glu)	29	28
Instability index:	50.42	50.70
Grand average of hydropathicity (GRAVY)	−1.379	−1.354

**Table 3 ijms-25-02951-t003:** The secondary protein structures before and after PIK3R1 mutation.

Secondary Protein Structure	Before Mutation	After Mutation
Alpha helix (Hh)	69.12%	61.76%
Random coil (Cc)	22.06%	29.41%

**Table 4 ijms-25-02951-t004:** The specific information of the 6-month-old sheep.

Sample	Live Weight (kg)	Carcass Weight (kg)	Dressing Percentages (%)	Ribeye Area (cm^2^)	Marbleization Grades
STH1	29.25	13.25	45.30	12.14	1.5
STH2	26.68	12.05	45.16	10.62	1.5
STH3	31.75	13.75	43.31	11.78	1.0
STH4	27.60	12.85	46.56	12.02	2.0
STH5	28.00	13.15	46.96	11.72	1.5
STH6	27.20	12.45	45.77	12.28	2.0
SFK × STH1	38.65	19.35	50.06	13.45	2.5
SFK × STH2	39.25	19.45	49.55	14.74	2.0
SFK × STH3	44.60	23.25	52.13	14.44	3.0
SFK × STH4	40.05	21.75	54.31	15.68	3.5
SFK × STH5	42.65	21.25	49.82	13.06	3.0
SFK × STH6	39.45	20.85	52.85	14.75	2.0

## Data Availability

The data presented in this study are available on request from the corresponding author.

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
