# Peer review of "Verification of Key Target Molecules for Intramuscular Fat Deposition and Screening of SNP Sites in Sheep from Small-Tail Han Sheep Breed and Its Cross with Suffolk"

_ijms, 2024, doi:10.3390/ijms25052951_

Round 1

Reviewer 1 Report

Comments and Suggestions for Authors

The authors analyzed the intramuscular fat (IMF) deposition mechanisms and related molecular makers in sheep by comparing small tail han sheep (STH) and F1 (STHp x Suffolk sheep).By transcriptome analysis, they found PIK3R1 and PPARA were candidate genes, as their expression was lower in F1 sheep. They also found a mutation site in exon 7 of the PIK3R1 gene, which is a potential genetic marker for IMF deposition.

Overall, the results are not clear enough to support the conclusions. Especially, as I pointed out below, the relationship between the SNPs and phenotypes are not described, even the genotype of the animals used in this study was not described. So, the importance of SNP found in this study is not clear.
In the material and methods section, the name and company are not described in many kits.
Some figures are not clear and difficult to read, so I think the international standards in data quality and data presentation are not met.

-The font size of “SFK×STH F1” is larger than other words in some places (L82, L111, and other lines), so please confirm and correct them.
-L153 and Fig.6: please delete “extremely” from these sentences (also L171).
-L215-229, Figures 9-10 and Tables 2-3: Are these contents really necessary? These are just predictions by the substitution of amino acids. The more important point is that the genotypes of STH and F1 used in this study (histology, RNA seq and IF (Fig. 1, 6, and 7).
-L297-299 Is the mutation described in ref [41] different from current study? Didn’t they get SNPs found in this study?
L319 “six sheep from the two groups”, does it mean 3 from STH and 3 from F1? The number of sheep analyzed in this study was not described.
L349 please indicate the name and company of the RNA extraction kit used in this study.
L350-354 also indicate the name and company of the kit used.
L384 CT should not be superscript. Aso indicate the kit name.
Figures 2-4: They are too small to read and no enlarged images were attached. I cannot read them.
Fig.6B What is “FPKM”? Is B also indicating gene expression? So, is 6B really needed?
Fig.7 What is negative control? I cannot distinguish signals in A-C. Are signals in A1&A2 nucleus? And what “APSAP” in D&E?
Fig.8 What are the right columns mean?
Table 2 What “Ho He Ne PIC X2(P)”?
References [28] and 26-38 are missing.

Author Response

Review1

The authors analyzed the intramuscular fat (IMF) deposition mechanisms and related molecular makers in sheep by comparing small tail han sheep (STH) and F1 (STHp x Suffolk sheep).By transcriptome analysis, they found PIK3R1 and PPARA were candidate genes, as their expression was lower in F1 sheep. They also found a mutation site in exon 7 of the PIK3R1 gene, which is a potential genetic marker for IMF deposition.

RE: I have received your revision suggestions and thank you very much for your careful guidance. I have made revisions to the manuscript based on your feedback, hoping that these modifications will make the work more perfect. If you have any other opinions or suggestions, please feel free to let me know. Regarding the genotype of the sample you proposed for this study, we would like to supplement it. The genotype and oil red staining positive signal area data of the sample selected for this study are shown in the following figure. As this data is not statistically significant, it was not mentioned in the article.

STH1

GA

0.3225

STH2

AA

2.619

STH3

GA

1.938

X1

GG

22.341

X2

AA

16.7715

X3

GA

7.721

1.The font size of “SFK×STH F1” is larger than other words in some places (L82, L111, and other lines), so please confirm and correct them.

RE: Thank you for your reminder. I have searched for similar errors throughout the text and made the necessary modifications. L19, L160, L19, L256, L283.

2.L153 and Fig.6: please delete “extremely” from these sentences (also L171).

RE:Thank you for your suggestion. We have made the necessary modifications based on it. L153, L171.

3.L215-229, Figures 9-10 and Tables 2-3: Are these contents really necessary? These are just predictions by the substitution of amino acids. The more important point is that the genotypes of STH and F1 used in this study (histology, RNA seq and IF (Fig. 1, 6, and 7).

RE: Thank you for your valuable suggestion. After careful consideration, we believe that this part of the content is necessary. The purpose of doing this section is to observe whether the reason for the inability of the two groups to deposit fat is due to a PIK3R1 mutation. After detecting a mistranslation mutation at this site, we want to once again prove that this mutation causes a change in protein structure, thereby altering its function. In fact, the existence of this part of the content also confirms that the mutation at this location causes changes in the primary structure amino acids of the protein, and the higher structure is also altered
4.L297-299 Is the mutation described in ref [41] different from current study? Didn’t they get SNPs found in this study?

RE: Thank you for your suggestion. In response to your question, the following explanation is provided: the mutation detected in the literature here is different from the mutation site in this article, but similarly, the literature also points out that the mutation in PIK3R1 is related to tail fat. This coincides with our viewpoint that the polymorphism of the PIK3R1 gene is closely related to fat deposition. However, due to the impact of the epidemic, this article did not conduct further research on the specific correlation between the two, which is also a shortcoming of this article.

5.L319 “six sheep from the two groups”, does it mean 3 from STH and 3 from F1? The number of sheep analyzed in this study was not described.

RE: Thank you for your feedback.We have provided detailed descriptions in the manuscript repeatedly. L327- L328and L331- L333

6.L349 please indicate the name and company of the RNA extraction kit used in this study.

RE:Thank you for your suggestion. We have made the necessary additions. L358- L368

7.L350-354 also indicate the name and company of the kit used.

RE:Thank you for your suggestion. We have made the necessary additions. L386
8.L384 CT should not be superscript. Aso indicate the kit name.

RE: Thank you for your suggestion. It has been revised and supplemented. L394

9.Figures 2-4: They are too small to read and no enlarged images were attached. I cannot read them.

RE: Thank you for your suggestion. I have uploaded the PDF format image in the attachment.
10.Fig.6B What is “FPKM”? Is B also indicating gene expression? So, is 6B really needed?

RE: Thank you for your suggestion, which has been explained in the manuscript. L155- L156 11.Fig.7 What is negative control? I cannot distinguish signals in A-C. Are signals in A1&A2 nucleus? And what “APSAP” in D&E?

RE: Thank you for your suggestion. This part may not appear obvious to the naked eye due to photography reasons. We have extracted the positive signal using Image j professional software, and APSAP refers to the density of the positive signal. L412- L413

12.Fig.8 What are the right columns mean?

RE: Thank you for your suggestion. It is mentioned in the methods section that a co localization analysis image is used for two types of positive signals,and has been revised accordingly in the manuscript. L427- L428

13.Table 2 What “Ho He Ne PIC X2(P)”?

RE: Thank you for your suggestion. The manuscript has been revised accordingly. L218- L219

14.References [28] and 26-38 are missing.

RE: Thank you for your suggestion. The manuscript has been revised accordingly. L546- L565

Reviewer 2 Report

Comments and Suggestions for Authors

The manuscript that has been submitted for review aims to explore the mechanism for еintramuscular fat deposition and the molecular markers in sheep.  Although a large number of studies are dedicated to this matter, the present one is original since it concerns also a local breed and its cross. The manuscript has a high scientific value, providing new and valuable information about the candidate genes and IMF deposition in this specific breed and cross, but also demonstrates high potential for the results to be implemented in practice.

A specific remark in my opinion can be made about the title of the manuscript. It needs to be more precise. The present title is too general and it remains unclear that the study is on sheep. In this regard, I think it would be better if the authors correct the title as Verification of Key Target Molecules for Intramuscular Fat  Deposition and Screening of SNP Sites in sheep from Small Tail Han sheep breed  and its cross with Suffolk.

The Introduction contains the necessary information about the state of the art and how the study is beyond the state of the art. However, the aim should be more clearly defined in the Introduction as it is done in the Abstract. Instead at the end of the Introduction, the authors have rather in details described what is done in the study. The paragraph should be kept but also the aim of the study should be more clearly presented as well.

The methodological approach for achieving the aim of the study is correct, however my main concern is about the number of the animals. In my opinion for these kind of studies at least 50 animals are required.

The material and method section is described in sufficient details. The methods applied are modern and guarantee reliable results.

The results are very clearly presented and also the number of tables and figures is adequate. The discussion is well made and the conclusions are sound.

Comments on the Quality of English Language

The English language is good, only a minor correction is needed: In the Abstract: Line 18: Histological staining revealed that the muscle fiber gap in the SFK × STH F1 was larger and higher IMF content. In my opinion should be: Histological staining revealed that the muscle fiber gap in the SFK × STH F1 was larger and the IMF content was higher.

Author Response

Review2

The manuscript that has been submitted for review aims to explore the mechanism for еintramuscular fat deposition and the molecular markers in sheep.  Although a large number of studies are dedicated to this matter, the present one is original since it concerns also a local breed and its cross. The manuscript has a high scientific value, providing new and valuable information about the candidate genes and IMF deposition in this specific breed and cross, but also demonstrates high potential for the results to be implemented in practice.

RE:We have carefully considered your feedback and made corresponding revisions to the entire text. We greatly appreciate your professional advice, which enables us to better provide high-quality services.

1.A specific remark in my opinion can be made about the title of the manuscript. It needs to be more precise. The present title is too general and it remains unclear that the study is on sheep. In this regard, I think it would be better if the authors correct the title as Verification of Key Target Molecules for Intramuscular Fat  Deposition and Screening of SNP Sites in sheep from Small Tail Han sheep breed  and its cross with Suffolk.

RE:Thank you very much for your valuable feedback. We have revised the manuscript title based on your suggestions. L2- L4

2.The Introduction contains the necessary information about the state of the art and how the study is beyond the state of the art. However, the aim should be more clearly defined in the Introduction as it is done in the Abstract. Instead at the end of the Introduction, the authors have rather in details described what is done in the study. The paragraph should be kept but also the aim of the study should be more clearly presented as well.

RE: Thank you again for your valuable feedback. We have made revisions to the preface section to make the purpose of the article more clear. L48- L49

3.The methodological approach for achieving the aim of the study is correct, however my main concern is about the number of the animals. In my opinion for these kind of studies at least 50 animals are required.

RE: Thank you for your suggestion. We have also carefully considered this part of the content, and in the end, there are a total of 55 samples in the entire population, which meet the criteria for SNP.

4.he English language is good, only a minor correction is needed: In the Abstract: Line 18: Histological staining revealed that the muscle fiber gap in the SFK × STH F1 was larger and higher IMF content. In my opinion should be: Histological staining revealed that the muscle fiber gap in the SFK × STH F1 was larger and the IMF content was higher.

RE: Thank you for your suggestion. We have made the necessary modifications based on it. L18- L20

Reviewer 3 Report

Comments and Suggestions for Authors

Traditional foods and unique raw materials are essential in many regions of China. Gansu is one of the number one sheep-producing and consuming provinces with several National Geographical Indication Protection Products.   Jingyuan lamb is one of the region's most famous dishes, prepared young lamb with extraordinary taste and tenderness. All these quality parameters are connected with intramuscular fat (IMF) contents. Some indigenous sheep, like the Tan breed, have high IMF. However, the Small Tail Han (STH) breed is known as a prolific lean breed since consumers' requests are quality, higher IMF containing lamb, the selection for it could be beneficial in this breed too.  

The authors investigated the potential candidate markers for higher fat-containing lamb meat selection in STH.  The title is clear; however, completing it with sheep will cover the topic more profoundly.

All parts of the Manuscript are very comprehensive and well-structured. The reader can easily follow the authors’ train of thought. After several filters, they described the molecular background of IMF deposition in sheep and the potential candidate genes (PIK3R1 and PPARA) regulating these processes in the experimental population. PIK3R1 could be a feasible marker for further selection in sheep.

Minor corrections are recommended:

L40 please rephrase the sentence.

L107-108 please rephrase the title of the figure (longest dorsal muscle is not correct).

L319-322 please add information about  the six-month-old lambs  (live weight, carcass weight, dressing percentages etc.)

Author Response

Review3

Traditional foods and unique raw materials are essential in many regions of China. Gansu is one of the number one sheep-producing and consuming provinces with several National Geographical Indication Protection Products.   Jingyuan lamb is one of the region's most famous dishes, prepared young lamb with extraordinary taste and tenderness. All these quality parameters are connected with intramuscular fat (IMF) contents. Some indigenous sheep, like the Tan breed, have high IMF. However, the Small Tail Han (STH) breed is known as a prolific lean breed since consumers' requests are quality, higher IMF containing lamb, the selection for it could be beneficial in this breed too.

RE:Thank you for your feedback. I have carefully studied your feedback and made modifications to the relevant content based on your suggestions.

1.The authors investigated the potential candidate markers for higher fat-containing lamb meat selection in STH.  The title is clear; however, completing it with sheep will cover the topic more profoundly.

RE:Thank you very much for your valuable feedback. We have revised the manuscript title based on your suggestions. L2- L4

2.L40 please rephrase the sentence.

RE: Thank you for your suggestion .We have made modifications based on your suggestions. L40- L41

3.L107-108 please rephrase the title of the figure (longest dorsal muscle is not correct).

RE: Thank you for your suggestion. We have made the necessary corrections to this error. L110- L112

4.L319-322 please add information about  the six-month-old lambs  (live weight, carcass weight, dressing percentages etc.)

RE: Thank you for your suggestion again, we have added the corresponding content based on your suggestion. L331- L333